# Deepening Inside the Pictorial Layers of Etruscan Sarcophagus of Hasti Afunei: An Innovative Micro-Sampling Technique for Raman/SERS Analyses

**DOI:** 10.3390/molecules24183403

**Published:** 2019-09-19

**Authors:** Rossella Gagliano Candela, Livia Lombardi, Alessandro Ciccola, Ilaria Serafini, Armandodoriano Bianco, Paolo Postorino, Lorella Pellegrino, Maurizio Bruno

**Affiliations:** 1STEBICEF Department, University of Palermo, Viale delle Scienze Ed. 17, 90128 Palermo, Italy; r.gaglianocandela88@gmail.com; 2Dipartimento di Chimica, Università degli Studi di Roma “La Sapienza”, Piazzale Aldo Moro 5, 00185 Rome, Italy; livia.lombardi@uniroma1.it (L.L.); alessandro.ciccola@uniroma1.it (A.C.); ilaria.serafini@uniroma1.it (I.S.); armandodoriano.bianco@uniroma1.it (A.B.); 3Dipartimento di Fisica, Università degli Studi di Roma “La Sapienza”, Piazzale Aldo Moro 5, 00185 Rome, Italy; paolo.postorino@uniroma1.it; 4Regional Center for the Design and Restoration of Cultural Heritage, Regional Department of Cultural Heritage and Sicilian Identity, Via dell’Arsenale 52, 90142 Palermo, Italy; pelegelino@gmail.com

**Keywords:** pictorial layer, Etruscan, sarcophagus, SERS analyses, Egyptian blue, madder lake, microsampling technique

## Abstract

The Hasti Afunei sarcophagus is a large Etruscan urn, made up of two chalky alabaster monoliths. Dated from the last quarter of the third century BC, it was found in 1826 in the small town of Chiusi (Tuscany- Il Colle place) by a landowner, Pietro Bonci Casuccini, who made it part of his private collection. The noble owner’s collection was sold in 1865 to the Royal Museum of Palermo (today under the name of Antonino Salinas Regional Archaeological Museum), where it is still displayed. The sarcophagus is characterized by a complex iconography that is meticulously illustrated through an excellent sculptural technique, despite having subjected to anthropic degradation and numerous restorative actions during the last century. During the restoration campaign carried out between 2016 and 2017, a targeted diagnostic campaign was carried out to identify the constituent materials of the artefact, the pigments employed and the executive technique, in order to get an overall picture of conservation status and conservative criticalities. In particular, this last intervention has allowed the use of the innovative micro-sampling technique, patented by the Cultural Heritage research group of Sapienza, in order to identify the employee of lake pigments through SERS analyses. Together with this analysis, Raman and NMR technique have completed the information requested by restorers, for what concerns the wax employed as protective layers, and allowed to rebuild the conservation history of the sarcophagus. In fact, together with the identification of red ocher and yellow ocher, carbon black, Egyptian blue and madder lake, pigments compatible with the historical period of the work, modern pigments (probably green Paris, chrome orange, barium yellow, blue phtalocyanine) have been recognized, attributable with not documented intervention during the eighteenth and twentieth centuries.

## 1. Introduction

The field of conservation of cultural heritage can nowadays easily be considered as one of the most challenging fields. The analytical approaches applied to cultural heritage works require highly sensitive, micro- or non-invasive techniques, and furthermore the application of several techniques, complementary to each other, to answer to the different questions concerning the historical and conservation issues. 

In particular, Raman Spectroscopy and its Surface Enhanced variant (SERS) have being appeared in the last decades as one of the most promising and efficiency techniques in such directions, allowing identification of pigments, analysis of micro-samples and characterization of natural dyes in artworks and ancient objects. Thus, the combination of the two approaches has extended the amount of achievable information [1,2]. The application of standard Raman spectroscopy for diagnostic purposes, in fact, is not always effective in characterization of coloring materials: if it usually results successful for the identification of inorganic mineral pigments, it could be affected by more drawbacks when the investigated sample presents a high content of organic material that is strongly fluorescent at the laser wavelength. This fact results in strong background interferences in the acquired spectrum, where the Raman signals are not easily observable. Typical examples of these issues with samples with high content of aged binder, adhesion of contaminants or, as occurs in several cases, by the presence of organic dyes. These colorant species, in fact, are usually strongly fluorescent under the laser and present in very low concentration; consequently, the acquisition of their characteristic Raman spectra from aged samples results really hard.

From this point of view, the application of SERS has represented a turning point in the analysis of art objects contaning organic dyes: the interaction of the dye molecules with the noble metal nanostructures used in SER spectroscopy, results in both fluorescence suppression and signal enhancement, which allow one to overcome the problems connected to the small amounts and highly fluorescent characteristics of organic lakes [1,2].

However, it is important to highlight that SERS analysis is based on the close interaction of analyte molecules with the metal nanoparticles: for this reason, the extraction of the dyes from the sample through acids, bases and chelating agents has usually represented a mandatory step for the certain identification [3]. Unfortunately, this pretreatment involves a sampling from the art-object, which could be risky and invasive, while the direct contact of the art object with an extracting solution could damage it. This set of problems has led to new approaches in the last years, which involve sampling through a gel matrix where metal nanoparticles are dispersed [4,5,6]: the polymeric substrate, loaded with water or the extracting solution, is put in contact with the pictorial film, with extraction of the dyes and consequent interaction with the nanostructures. This makes it possible to acquire SER spectra directly on the gel substrate, reducing the risk of damaging the artwork while simultaneously providing high quality data. 

The case study reported in this paper is a clear example of this new diagnostic approach. The presented results, in fact, not only show how Raman and SERS analysis can be successfully employed and provide answers to conservation issues, already demonstrated by several other works in different application cases [7,8,9,10,11,12,13], but furthermore reports the use of an innovative micro-sampling technique (GT-SERS), patented by La Sapienza and the Cultural Heritage group (Chemistry Department), which overcomes the problem of micro sampling, extraction and SERS diagnostics: a protocol generally used for the identification of natural dyes in art objects [1,12,14,15,16]. The patented micro-sampling technique overcomes the problems connected to sample treatment, allowing traditional Raman and SER spectra acquisition directly on micro-fragments mechanically sampled from the artwork, with no use of solvents.

The aim of this investigation is to gain an understanding of the pigments employed in a Hellenistic sarcophagus, with the main purpose of determining if it is a false or not. The sarcophagus in question was discovered in the small town of Chiusi, located between Tuscany and Umbria (Italy), in the locality called “il Colle”, belonging to the noble landowner Pietro Bonci Casucci [17]. In 1826 a hypogean tomb was discovered and inside it, the large Hellenistic sarcophagus (dated to the last quarter of the 3rd century BC). It is composed of two large monoliths, lid and box, thoroughly sculpted, decorated and painted (Figure 1a).

The sarcophagus is thought as a kline (lid) on which a woman is represented and for her the name of Hasti Afunei is indicated. The sarcophagus is headless and without arms (the tomb was desecrated and looted before 1826), lying down on a triclinium, richly dressed and jeweled. From the processing of the constituent material, chalky alabaster, it is possible to perceive the materiality of the different fabrics of the dress or the jewels. On the box there is a scene of a funeral relief, in which the deceased says goodbye for the last time to her family.

During the last restoration of the sarcophagus (2016–2017) a campaign was carried out in order to identify the constituent materials. Before of the intervention, the sarcophagus was affected by various degradations and alterations. For these reasons it was not possible to achieve a correct perception of the conservation status of the artwork. Once identified the pigments employed, the authenticity of the work has been verified, distinguishing also the previous restoration interventions. In fact, pigments incompatible with the historical period of the archaeological artwork and the presence of surface protections have been identified and related to different periods and previous restoration activities.

Thanks to this information, a tentative 3D color reconstruction has been made, once identified all the original pigments employed (Figure 1a,b). This work was based on evaluating both microscopically and macroscopically the entire surface in the artifact by diagnostic analyses (XRF mapping and Raman punctual analysis), archaeological study, and comparison with other similar Etruscan works of art making it possible to imagine the original state of the artifact.

## 2. Results 

### 2.1. Surface Protection Products

On the surface of the artwork, two protective coating agents have been found, probably applied in different historical periods. By NMR spectroscopy the presence of paraffins was observed (see the Appendix A for further detail).

### 2.2. Pictorial Layers

The entire surface has been observed macroscopically, in order to map the color areas. A complete removal of coating agents has been performed in those areas in which further analyses of pigments would be pursued. In some cases, a scalpel sampling has been also performed where colored areas were more extensive. GT-SERS has been performed in those areas in which pictorial layers appears more limited. The sampling points are shown in Figure 2 and Figure 3. 

#### 2.2.1. Standard Raman Spectroscopy

The Raman analysis was performed on nine samples, in particular on blue (six samples), red (two samples) and purple parts (one sample) of the art object. These areas represent those with main color residues, so they allowed a less invasive sampling with scalpel, providing information about the main pigments present on the object.

##### Blue Pigments

The sarcophagus presents different painted details. A blue paint layer is still visible, both on the lid and in the case. Preliminary Raman analysis with portable devices did not allow the aquisition of any remarkable spectra due to the high fluorescence widely spread all over the sculpture. Micro-samples of the blue paint layer were taken mostly from the lid of the sarcophagus: from the woman’s belt and dress, and from the pillow and the triclinium’s mattress. From the case, one sample from the background and another one from the clothes of one character were examined. The Raman spectra obtained were generally consistent: all of them presented characteristic peaks at 431 and 1086 cm^−1^, which can be respectively assigned to bridging O breathing and to SiO stretching vibrations [18,19,20,21,22] (Figure 4; sample D1 is shown as indicative of other points). Moreover, two peaks, at 111 and 139 cm^−1^, respectively, are also present in every spectrum: this set of peaks is indicative of Egyptian blue (CaCuSi_4_O_10_), one of the most ancient synthetic pigments, used also by the Etruscan people [22].

##### Red Pigments

Two samples were collected with the scalpel, one from the dress of the woman and another one from a side red spot on the harm; this second spot was optically attributable to successive restorations and not original, appearing as a varnish. In both the collected Raman spectra (Figure 5) clear signals of hematite (Fe_2_O_3_) were observed. In particular, for both the samples, characteristic peaks at 220 for A_1g_ mode, 291 and 407 cm^−1^ for E_g_ modes are observable. Furthermore, for the red spot sample, Raman spectrum presents a higher signal-to-noise ratio and also minor intensity peaks at 246 (E_g_), 493 (A_1g_), 610 (E_g_) are also observable. The final confirmation is represented by a broad band at 1320 cm^−1^, attributable to two-magnon scattering (excitation of collective spin movement) [15,16,23,24]. The presence of hematite is indicative of use of a red ochre as pigment.

##### Purple Pigment

The second character from the left of the case, presents, on the dress, some fragmentary decorations of purple color. This pigment was taken to be identified by micro-Raman spectroscopy. By magnifications under the microscope it was observed that the pigment was a mixture of two different ones: one blue and one red. The blue one shows the characteristic spectrum of Egyptian Blue [18,19,20,21,22] and the red pigment was identified as a red ochre, because of its content of hematite (intense peaks at 220 for A_1g_ mode; at 243, 291, 410 and 610 cm^−1^ for Eg modes) [18,19,23,24].

#### 2.2.2. Gel-Transfer Surface Enhanced Raman Spectroscopy (GT-SERS)

The sarcophagus presents other less extended parts with pictorial residues. Taking into account the different colors and the difficulties in performing a micro-invasive sampling in these areas, the gel-transfer micro-invasive sampling method was applied in order to evaluate the composition of the different colors.

##### Blue Pigments

The gel-transfer device was used to sample a blue fragment from the chest of the dress of the first character Culsu, on the case of the sarcophagus. In this sample phthalocyanine blue (PB15) [25] was identified (Figure 6) by comparison with an internal reference. Characteristic peaks are observable at 680 cm^−1^ (C-C, C-N and C-N-C benzene deformation); 747 cm^−1^ (Cu-N, C-N-C and C-C-H); 952 cm^−1^ (N-Cu-N and C-C-N); 1106 cm^−1^ (C-N-C and Cu-N); 1142 cm^−1^ (N-C-N and C-C-H); 1304 cm^−1^ (isoindole deformation); 1451 cm^−1^ (C-C); 1527 cm^−1^ (C-N) [26]. This pigment is a copper coordination compound which was synthesized in the first decade of the XXth century, so its presence must be attributed to a prior restoration.

##### Brown Pigments

The hairs of the characters, present on the case (Figure 4), are brown colored. By applying the sampling device, very small particles (as reported in the Experimental section) were mechanically extracted. The Raman spectra collected from the micro-samples trapped inside the gel matrix allowed the identification of hematite (Fe_2_O_3_, characteristic peaks at 220, 291 and 407 cm^−1^) [18,19,23,24] and goethite (Fe(OH)_3_, peaks at 300 and 386 cm^−1^) [24] (Figure 7a,b). These minerals are indicative of the use of ochres as pigments, likely mixed to obtain a dark color that tends to warmer tones. The variation of the SERS spectra referring to standard Raman spectra of reference compound could be attributed to intensity variability of SERS signals due to local interaction with nanoparticles. 

##### Black Pigment

The case of the sarcophagus presents a frame in which the names of characters are written with a black paint (epigraph). By mean of this inscription it was possible to also know the story of the Etruscan Afuna family. The epigraph is fragmentary in some portions. It seems to have been rewritten during previous restorations or perhaps copied the same day of the discovery in 1826 [17]. The Raman spectra collected from the micro-samples trapped inside the gel matrix allowed the identification of carbon black from the two broad bands at 1320 and 1585 cm^−1^ (Figure 8) [18,19]. 

##### Green Pigment

The second figure from the left on the case is the one with the best preserved colors. The character is a demon and carries a big green key under his arm which is symbolic of the afterlife reign (Figure 9). The Raman spectra collected from the micro-samples trapped inside the gel matrix presented a lower signal to noise ratio and higher interference from gel substrate signals. On the basis of literature data, a tentative attribution suggests the presence of three different pigments: emerald green (Cu (CH_3_COO)_2_ 3 Cu(AsO_2_)_2_– peaks at 430, 490, 534, 951 cm^−1^), chrome orange (PbCrO_4_·PbO peak at 346 and 828 cm^−1^) and barium yellow (BaCrO_4_– peaks at 861 and 901 cm^- 1^) (Figure 7) [15,16]. Further analyses, for instance elemental ones, would be necessary for sure confirmation of the assigned species.

##### Pink Pigment

The woman lying on the lid of the sarcophagus wears a blue dress with some pink areas (Figure 10). In this case, when a portable Raman instrument was used, a very strong fluorescence was observed, which was also higher in comparison to the other color pictorial layers. This suggested that, in this area, fluorescence could be not attributable only to the binder and contaminants, but it could be related to the presence of an organic lake, taking into account the color. The application of the micro-sampling protocol on gel matrix [16] confirmed this hypothesis. Raman spectra acquisition resulted in signals characteristic of madder lake: in particular, the high intensity peaks at 1294 cm^−1^ can be attributed to anthraquinone molecule C-C stretching and C-OH bending modes, while the signal at 1325 cm^−1^ is typical of madder dyes, and attributable to the presence of alizarin and purpurin. The peak at 1553 cm^−1^ is attributable to C-C stretching mode, while the peak at 1630 cm^−1^ to C-C and C=O bonds [27]. 

## 3. Discussion

The possibility of determining the composition, in terms of materials used for the realization of an artwork, is necessary for the curator and conservator, in order to answer historical questions (which materials were used at the time, what were the commercial routes, etc.). Moreover, it can also answer an important question, maybe the first where any artwork is discovered: whether the work of art has been manipulated or not and if it is actually a fake.

In this context, the possibility of performing micro-sampling is crucial, taking into account the value of artwork, which does not allow one to take large samples, and the difficulties that can occur during sampling, especially on sculptural works, with curved, rough surfaces.

The described case study represents a clear example of the crucial role of micro-sampling in providing an extended amount of information. In fact, for the abovementioned art object, Raman analysis with a portable instrument was not successful due to the high fluorescence background all over the surface. Because of this phenomenon, sampling from the art object was considered in order to deepen the investigation of the pictorial residues. The Raman analysis of fragments sampled with a scalpel resulted effective, allowing confirmation of the use of Egyptian blue for blue portions, hematite-rich ochre for red ones and a mixture of these pigments for a purple one. Unfortunately, traditional sampling with a scalpel was possible only in some more extended areas, whereas it could not be applied to other smaller portions that showed other colours. For this reason, the gel-transfer micro-sampling method was used. This technique, developed by Lombardi et al., is characterized by micro-invasivity and allowed to carry out the micro-sampling in the central area of and in the main figures of the paintings, where other sampling methodologies could not be employed. In this case, no addition of organic solvent to the patented SERS-pen was necessary. The analyses performed on the Ag-gel matrix, in fact, gave important results which extended the informative content of the diagnostic campaign. 

For what concerns the brown parts, the identified pigments—hematite and goethite—are compatible with the Etruscan period, although they are still used today. The two minerals could be present together in an earth pigment, or, alternatively, they could have been mixed to obtain a dark color that tends to warm tones. The black pigment detected in the writing is carbon black. Both ochres and carbon black are present as pigments in ancient paintings as well as contemporary art material, so it is not possible to consider them specifically as original materials, however, they are consistent with Etruscan age paintings [28].

The green pigments, on the other hand, are related to three different species: emerald green (Cu (CH_3_COO)_2_·3Cu(AsO_2_)_2_, chrome orange and barium yellow (Figure 7) [15,16]. These pigments are not attributable to an ancient period, so they must be the result of subsequent restorations. In particular, because they were no longer used after around 1900 because their toxicity, their presence could be related to the 19th century pictorial retouching performed by Vincenzo Monni during the restoration following the initial discovery of the work.

The use of gel-transfer sampling was also useful to obtain complementary information to a standard Raman analysis. In fact, all the standard Raman spectra were consistent for the identification of Egyptian blue; nevertheless, the spectrum acquired on the fragment entrapped in the gel matrix showed all the peaks of a synthetic organic pigment, the phthalocyanine-based PB15. This allowed the identification of a painted area which had been subjected to restoration and must be considered as not original; moreover, because of the recent origin of the pigment found, it is possible to identify the mentioned restoration as the 1950 treatment performed by Rosario Forizisi at the Regional Archaeological Museum A. Salinas of Palermo.

Finally, the analysis of the pink colour has allowed us to individuate the presence of a lake pigment, which was identified as madder lake [8,9]. The dress on the chest probably had decorations of different colors. Perhaps the artist had decorated the dress with pink, or perhaps this rose was over the tones of Egyptian blue to create an iridescent effect. It is highly probable that the present lake is original and it is not a synthetic one obtained from pure alizarin: in fact, in the madder lake found on the artifact there are none of the untypical components (chrome, iron, tin) of “modern” manufacturing of the pigment created by George Fields, dating back to the early XIXth century. The madder lake in an alum and alkali matrix (if applied in antique restorations) would never have lasted to the present day. It is an organic dye that is not very stable to photo-oxidation phenomena. The madder lake found in the artifact has both alizarin and purpurin. The last one was removed from the industrial recipe in 1827 (one year after the discovery of the sarcophagus) to produce the colour ‘genuine rose garanza’ and from this date on the pigment wass marketed only as synthetic alizarin complexed with alum or other metal ions [28]. Real madder dyes, on the other hand, were widespread in antiquity and madder plants were cultivated as a dyestuff source since antiquity in Central Asia, South Asia and Egypt, where it was grown as early as 1500 BC. Clothes dyed with madder root dye were found in the tomb of the Pharaoh Tutankhamun and on an Egyptian tomb painting from the Graeco-Roman period, diluted with gypsum to produce a pink color, according to the reported iconography [25,29]. Like the case of Egyptian blue, the same technique was used in Etruscan culture with madder lake. This people, like the Egyptians, had the technology to stabilize an organic dye in an inorganic matrix. Lake pigment was incorporated into plaster whereas Egyptian blue was incorporated into a clay matrix. The use of madder lake with a chalky matrix for the colouring of a chalky alabaster sarcophagus is perfect.

For these reasons, the identification of the madder lake as well as the Egyptian blue is an important finding since it allows to hypothesize the originality of this paintings layer on the sarcophagus. 

This set of identification results highlights the potential of our micro-sampling technique, which is really versatile and allows acquiring Raman spectra even if a sample is affected by strong fluorescence. In fact, the protocol allows sampling small fragments of pictorial film which can be easily analysed with a micro-Raman spectrometer. When the fluorescence is not too strong, it is possible to collect Raman spectra under idoneous measurement conditions. This approach has been followed also for several pigments collected from the sarcophagus (blue, red, purple, black and green layers). When the fluorescence is remarkably high, instead, the presence of Ag nanoparticles inside the gel matrix results in a typical Surface Enhancement effect, which is useful to characterize highly fluorescent species, for instance organic lakes; in the case of the ancient sarcophagus, this resulted useful to identify the presence of madder lake in the pink layer. Clearly, in a case study, it is not possible to discriminate if the obtained spectrum is a standard Raman one or if it presents Surface Enhancement of Raman signal (the colloid species, in fact, are homogeneously distributed in the matrix, as observable from the presence of the characteristic signals of Ag nanoparticles at around 200 cm^−1^, present in all the spectra acquired on the gel substrate). However, from the application point of view, this offers the possibility of identifying a wider set of colorant species without preliminary information. Moreover, the immobilization of the samples in the gel allows handling them easily and performing further analysis. In comparison to other gel substrates used in the literature [6,30], the present methodology can be applied without any solvent as extracting agent: the sampling is prevalently mechanical, so the pre-treatment of the art-object surface with acid or bases was avoided. This contributes to the versatility of the technique: the direct application of the sampling device does not require preliminary analysis for the individuation of area which require extraction protocols, also potentially damaging for the artwork substrate (e.g., in this case, an acid extraction of the dye would not be idoneous for the alabaster substrate). It should be underlined that, in some cases (the yellow and green particles), the identification of inorganic material in the gel matrix was not easy because of intrinsic SERS variability and interference of gel substrate signals. This aspect should be further studied in future studies related to database construction. 

## 4. Experimental

### 4.1. Microsampling Technique for Gel-Transfer Surface Enhanced Raman Spectroscopy (GT-SERS):

The selected sampling kit was chosen because it allows micro-sampling on surfaces of different typologies (horizontal and vertical, rough and smooth) in a simple and safe way [11]. This tool is based on direct contact between a gel matrix, that has circular shape with a diameter of 0.5 cm, and the artwork and allows sampling of micro-fragments of the pictorial layer by mechanical adhesion. These micro-fragments are characterized by an average diameter of 10–12 μm (these data are confirmed by statistical analyses conducted for the development of the patented method and reported in the patent text). The sampling can be performed either by a mechanical trapping of the sample within the gelatinous matrix, by direct contact withwater present in the aliquot of rigid gel or by micro-extraction with a suitable organic solvent or a mixture of solvents placed on the surface of the gel. The protocol to carry out sampling using the kit devices is subdivided into the following steps:Place the gel in contact with the surface to be samples; addition of 1–2 µL of organic solvent (ethanol, methanol, acetone, hexane, etc.) or solvent mixtures could be necessary depending the nature of the substance to be sampled.Hold in contact for enough time (in the range of 1–30 s). This time range should be sufficient for all the cases, even if some variations could occur depending on several factors.After the alloted time, remove the gel matrix from the surface.Cut the portion of Ag-gel matrix containing the samples and place it on one of the supports provided in the kit. Sampling effectiveness can be checked through a portable USB optical microscope.Leave the Ag-gel matrix portions to dry in the sample holder in horizontal position. After drying (for 2–3 h) they will adhere to a glass slide. Portions are then immobilized and can be moved without any risk.Analysis in the laboratory can be performed on the microsample using a Raman micro-spectrometer, even after a long time. After Raman and SER spectroscopy analysis, the sample can be re-extracted from the gel matrix. This allows one to carry out other investigations (e.g.,: HPLC, mass spectrometry) for cross-validated analysis. This is possible because the characteristic insolubility of the gel.

### 4.2. Raman/SERS Analyses

The diagnostic campaign started with in situ Raman measurements carried out with a portable Raman HE instrument (High Efficiency Raman Analyzer, Horiba, Palaiseau, France) equipped with a 785 nm laser. These measurements did not allow the registration of spectra useful for determining the composition of the pictorial layers due to the intense florescence background exhibited by the surface of the work. This intense fluorescence was attributed to surface treatments based on organic substances that were subsequently identified through NMR analysis. Since non-invasive analyses did not allow the identification of pigments present on the sarcophagus, samples were collected both through classical scalpel sampling method and through an innovative patented gel-transfer micro-invasive sampling method [13]. Samples were taken both from the lid and the box in order to determinate if they were original and of the same composition. The Raman analyses of the collected samples were conducted through a Horiba Jobin-Yvon LabRam Confocal Microscope, equipped with 632 nm laser (experimental conditions: laser intensity 10%, microscope objective 100×, collection time 1 s, 100 accumulations)

### 4.3. NMR Analyses

NMR spectra were performed on three surface samples of the sarcophagus to determine the composition of the organic material. The samples were suspended in 600 μL of CDCl_3_. One-dimensional ^1^H spectra were obtained through 128 scans for each sample on a Bruker Avance III 400 spectrometer (Bruker Spectrospin, Karlsruhe, Germany) under the following conditions: 9.4 T at 298K, acquisition of the given 64K points FID, with a spectral window of 15 ppm, 128 scans and an intersection time of 10.28 s, for a total time of 15 s between consecutive scans.

## 5. Conclusions

By means of the Raman and SERS analyses it was possible to determine the presence of original pictorial drafting and other traces of alleged pictorial interventions during the XIXth and XXth centuries. The presence of large original paint layers that have survived over time and the vicissitudes of the artwork is certainly a matter of considerable importance. The state of conservation, after 2300 years, of pigments such as ochres, blue Egyptian and madder lake on chalky alabaster, not only involves a factor of preciousness of the work, but also determines the *modus operandi* and the artistic technique used by the Etruscans in the decoration of large urns made of alabaster, a technique similar to that used for the urns in travertine and in the so-called fetid stone. The scientific examination of the artwork was fundamental for the determination of the restoration interventions that the sarcophagus has undergone throughout its history and which are historically not fully documented. The use of many analytical techniques had helped us know the materials used for the realization of the archaeological artefact and has guided the latest restoration treatment (2017) during cleaning, consolidation and integration.

## Figures and Tables

**Figure 1 molecules-24-03403-f001:**
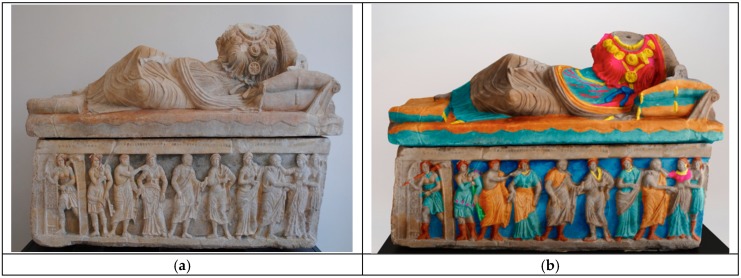
Hasti Afunei’s sarcophagus, after restoration (**a**), 3D reconstruction of the Hasti Afunei sarcophagus by the Arch. Alessandro Navarra (**b**).

**Figure 2 molecules-24-03403-f002:**
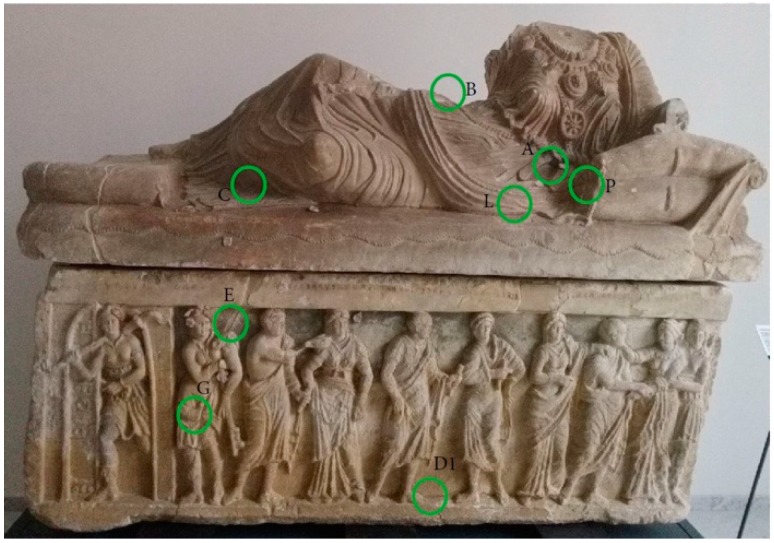
Scalpel sampling points.

**Figure 3 molecules-24-03403-f003:**
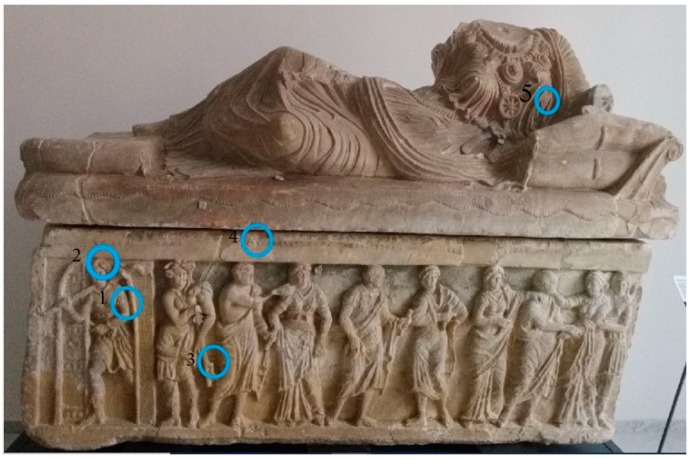
GT-SERS sampling points.

**Figure 4 molecules-24-03403-f004:**
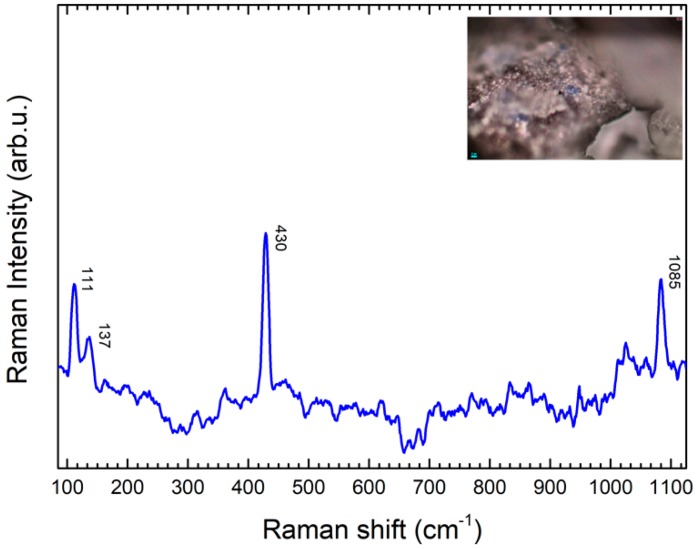
Raman spectrum obtained on a scalpel sample from the D1 area, showing the characteristic peaks of Egyptian Blue.

**Figure 5 molecules-24-03403-f005:**
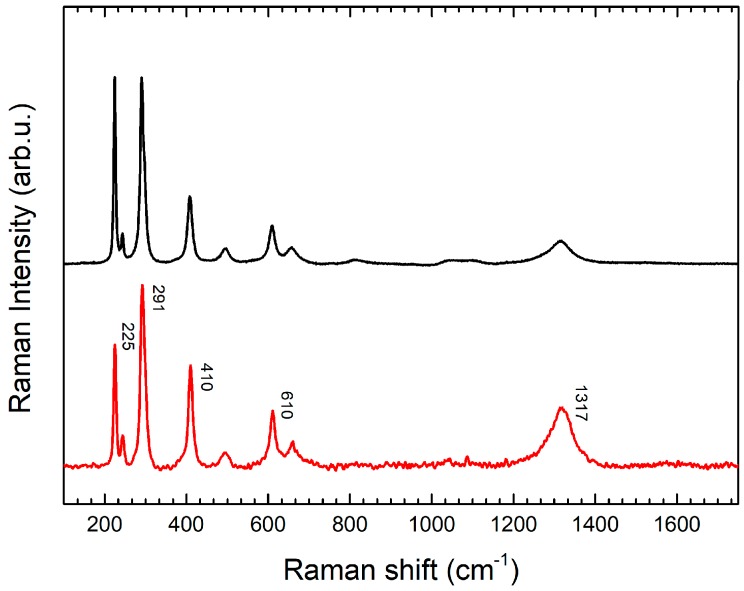
Raman spectrum (red line) of sample of red vanish, widespread all over the top of the sculpture, compared with the Raman spectrum of a hematite reference obtained with the same instrument (black line).

**Figure 6 molecules-24-03403-f006:**
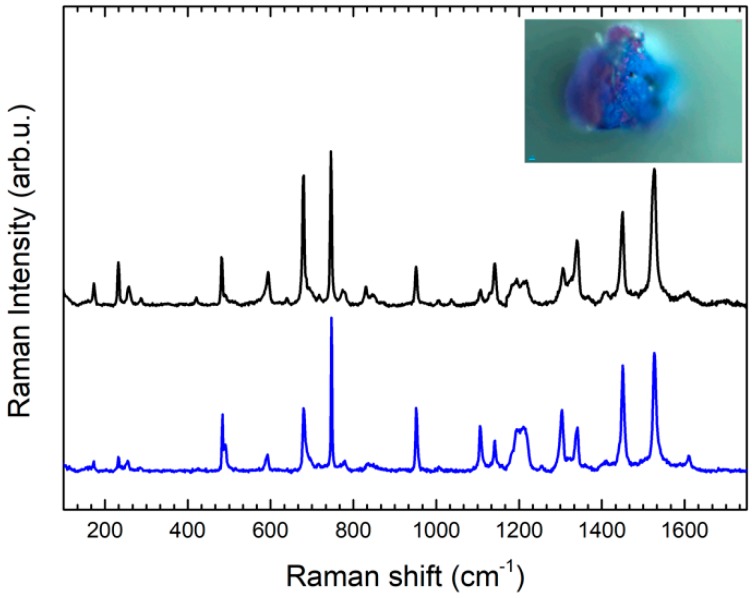
SERS spectrum (blue line) of sample collected with GT-SERS of blue pigments, 1 area, compared with Raman spectrum obtained with the same instrument of PB15:1 reference (black line).

**Figure 7 molecules-24-03403-f007:**
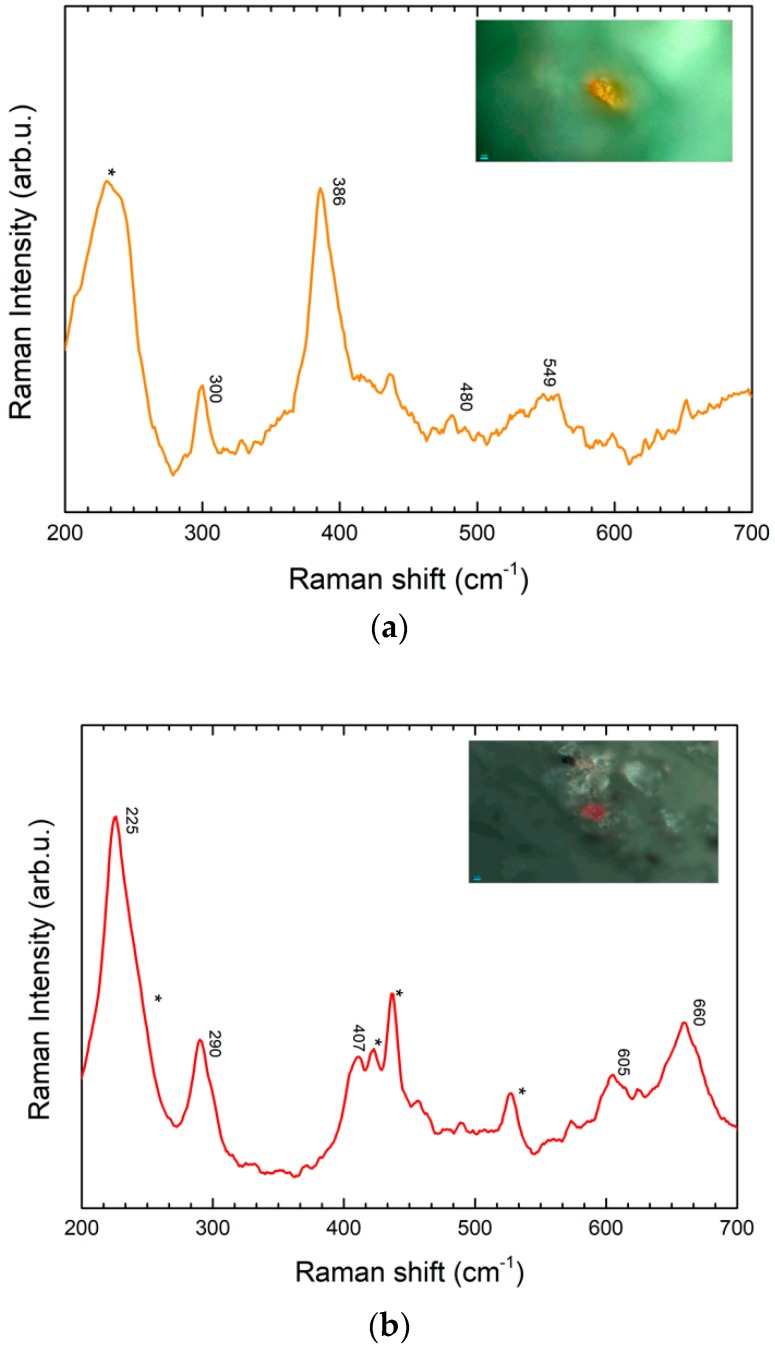
(**a**) SERS spectrum (orange line) of sample collected with GT-SERS of a yellow particle from area 2 with characteristic peaks of goethite. (**b**) SERS spectrum (red line) of a sample collected with GT-SERS of a yellow particle from area 2, with characteristic peaks of hematite. Asterisks are indicative of gel substrate peaks.

**Figure 8 molecules-24-03403-f008:**
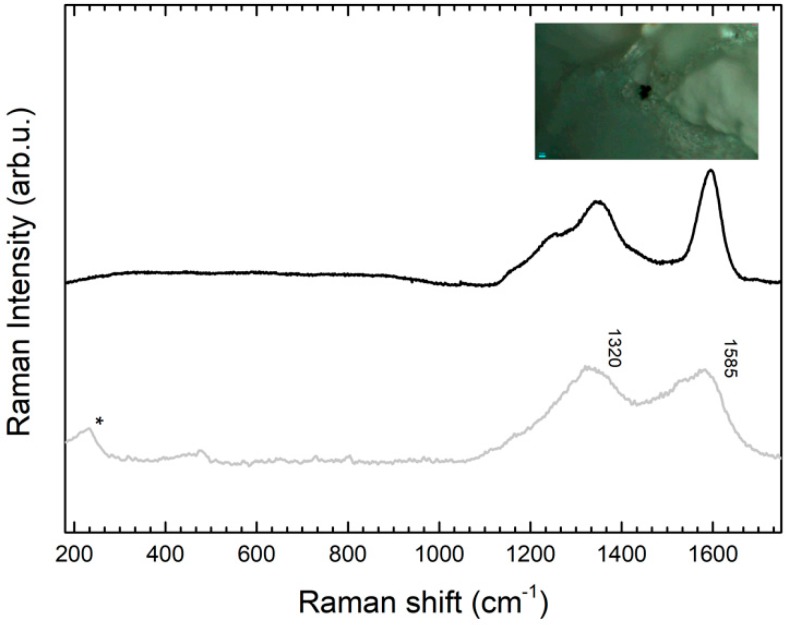
SERS spectrum (grey line) of a sample of the black pigment from area 4 collected with GT-SERS, compared with the Raman spectrum of a carbon black reference sample obtained with the same instrument (black line).

**Figure 9 molecules-24-03403-f009:**
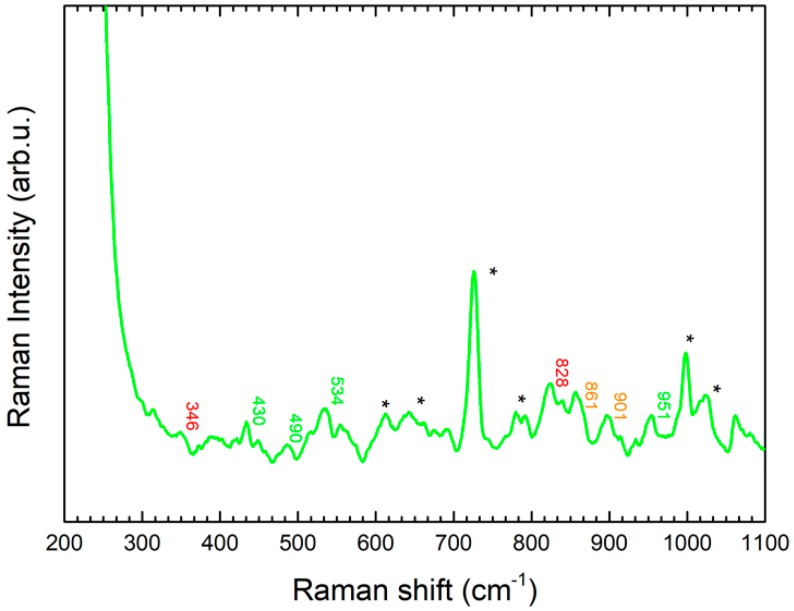
SERS spectrum of a green pigment sample collected from area 3 area with GT-SERS, showing characteristic peaks of emerald green (Green text), chrome orange (red text) and barium yellow (yellow text). Asterisks are indicative of gel substrate peaks.

**Figure 10 molecules-24-03403-f010:**
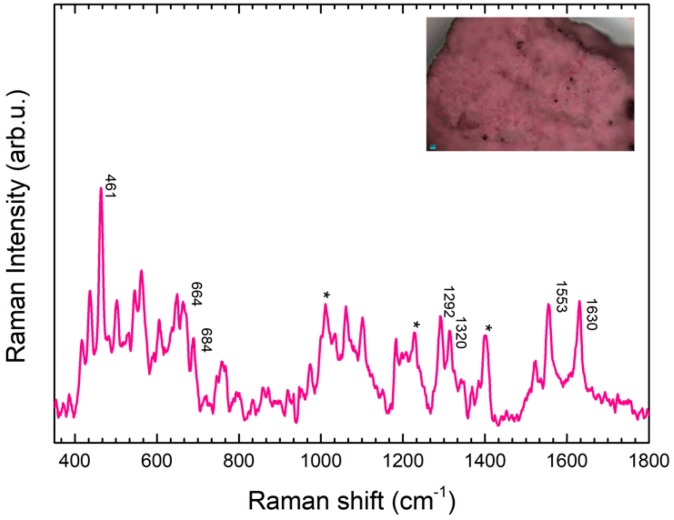
SERS spectrum of a pink pigment sample, collected from area 5 by GT-SERS, showing characteristic peaks of madder lake. Asterisks are indicative of gel substrate peaks.

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
