# Peer review of "Deepening Inside the Pictorial Layers of Etruscan Sarcophagus of Hasti Afunei: An Innovative Micro-Sampling Technique for Raman/SERS Analyses"

_molecules, 2019, doi:10.3390/molecules24183403_

Round 1

Reviewer 1 Report

The paper "Deepening inside the pictorial layers of Etruscan sarcophagus of Hasti Afunei: an innovative microsampling technique for Raman/SERS analyses ", molecules-570765, fits into "Molecule" topics as an application of chemistry in cultural heritage, but it is not acceptable for publication in its present form

There are some changes/corrections I suggest so that all the readers could best understand the information given by the authors.

A)Authors must discuss their molecular band assignments and display whenever possible references acquired in the same conditions.

Importantly, they must thoroughly discuss how confident they are with the pigment attributions they propose. For example, in the case of the identification of a madder lake, I suppose that to prepare a sample to be analysed by Surface-enhanced Raman scattering they will have destroyed the Al3+-complex, i.e, the lake; so, they will be analysing anthraquinone dye(s)? if so, they should state so and provide a molecular structure for the molecule(s). In brief, they must discuss the experimental evidence they have acquired to conclude it is a pigment lake.

B)The quality of the figures of Raman spectra must be improved to be published in a high quality journal as Molecule; presently, they are displayed inside a box, in a very small size which makes difficult to assess the spectral information: band numbers and overall bands are difficult to read

C)In the manuscript, it is not clear which samples where collected with a scalpel and which with the patented method. Fundamental information is missing concerning the sample size and, in the case of the gel application, the area of contact.  It will be also important to discuss the safeness of the method; e.g., concerning residues left in the work of art.

For the application of the method, in the reference list, authors refer to the patent [11]. So, one may conclude that this will be the first work published applying this extraction procedure? If so, authors should compare it with other methods, published in the literature, using gel to extract dyes for SERS; e.g., the techniques tested by Leona, Casadio, Pozzi.

It is not clear why the "wax" coating applied in the work of art was not analysed by microFTIR? it would have allow to analyse a higher number of samples, preserving them for further testing if necessary. Again, authors must provide information on the sample size, and if possible the sample mass. How much is it necessary to obtain an NMR spectra, and how does it compare with the amount required for a microFTIR analysis?

D)Another important point is related with the "3D colour reconstruction proposed in Fig 1B"

From the data presented and discussed in their manuscript, it will be not possible to propose a colour reconstruction. It is not possible to predict a final colour through a pigment; pigments are certainly important for determining colour, but colour built-up & technique are also fundamental for the final colour. Colours were not "uniformly sprayed" in antiquity.

Possibly, in the legend for Figure 1B and in the text, authors could propose colour areas for blue, pink, etc. These areas would be indicative of the generic hue, but should not be considered a colour reconstruction.

However, to be able to provide these generic colour areas, more data than what is provided by micro-sampling and discussed in the paper, is necessary. Possibly, authors had used XRF to indicate areas of colour? and they did not describe it in the manuscript submitted

Other points, related to A), that is is fundamental to fully address are highlighted bellow

1) For Egyptian blue, please describe band assignments (in the text or in a table)

2) Please provide spectra of reference samples, acquired in the same conditions, for goethite and hematite

3) Red pigment in purple, please provide band assignments

4) Green pigment; it is fundamental to provide spectra of reference samples in Fig 7 and it is necessary to discuss in detail its composition; please provide band assignments

5) Pink pigment, please provide reference compound, peak assignments and discuss the accuracy of your prediction in which concerns the identification of a madder lake pigment

Please discuss, why you discard that it could also be a restoration pigment? Considering that in until recently, at least, Winsor & Newton was selling a "genuine" madder lake pigment.

Author Response

Reviewer #1:

Comments to the Author

The paper "Deepening inside the pictorial layers of Etruscan sarcophagus of Hasti Afunei: an innovative microsampling technique for Raman/SERS analyses ", molecules-570765, fits into "Molecule" topics as an application of chemistry in cultural heritage, but it is not acceptable for publication in its present form

There are some changes/corrections I suggest so that all the readers could best understand the information given by the authors.

Authors must discuss their molecular band assignments and display whenever possible references acquired in the same conditions.

Importantly, they must thoroughly discuss how confident they are with the pigment attributions they propose. For example, in the case of the identification of a madder lake, I suppose that to prepare a sample to be analysed by Surface-enhanced Raman scattering they will have destroyed the Al3+-complex, i.e, the lake; so, they will be analysing anthraquinone dye(s)? if so, they should state so and provide a molecular structure for the molecule(s). In brief, they must discuss the experimental evidence they have acquired to conclude it is a pigment lake.

ANSWER: in literature, SERS spectra of lake pigments and madder dyes after the breakdown of complex are reported; for example, it is possible to see in Historical Organic Dyes: a Surface Enhanced Raman Scattering (SERS) spectral database on Ag Lee-Meisel colloid aggregated by NaClO4 by Silvia Bruni, Vittoria Guglielmi and Federica Pozzi in Journal of Raman Spectroscopy 2011, 42, 1267-1281. In this paper, the comparison between the SERS spectra of madder dyes upon extraction and the madder lake are reported.

For what concerns molecular structures of anthraquinone dyes, the complete characterization of dyes composition is available in different publications, such as Application of Surface Enhanced Raman Scattering technique to the ultrasensitive identification of natural dyes in works of art, by Marco Leona, Jens Stenger and Elena Ferloni in Journal of Raman Spectroscopy, 2006, 37, 981-992.

The quality of the figures of Raman spectra must be improved to be published in a high quality journal as Molecule; presently, they are displayed inside a box, in a very small size which makes difficult to assess the spectral information: band numbers and overall bands are difficult to read

ANSWER: We modified the figures in order to improve their general quality and make spectral features clearer.

In the manuscript, it is not clear which samples where collected with a scalpel and which with the patented method. Fundamental information is missing concerning the sample size and, in the case of the gel application, the area of contact.  It will be also important to discuss the safeness of the method; e.g., concerning residues left in the work of art.

For the application of the method, in the reference list, authors refer to the patent [11]. So, one may conclude that this will be the first work published applying this extraction procedure? If so, authors should compare it with other methods, published in the literature, using gel to extract dyes for SERS; e.g., the techniques tested by Leona, Casadio, Pozzi.

ANSWER: As reported in the patent, which is available in every patent library, the Ag-gel consists of a mechanical pencil that has inside the polymeric species in which are dispersed particles of silver and other chelating compounds. The contact of this gel with the surface of the work of art leaves no harmful residue. This was evaluated by different techniques in the design of the kit (See the patent text) and, in our case studies, it was also confirmed by microscopy observation.

Furthermore, the paper has been prepared to be submitted a special issue concerning Natural Dyes and, in this case, the approach to a microsampling technique has allowed to recognize madder lake. However, a complete description of the kit is beyond the scope of these paper and we refer to the patent in which every information is included. In fact, public patent represents a publication itself.

It is not clear why the "wax" coating applied in the work of art was not analysed by microFTIR? it would have allow to analyse a higher number of samples, preserving them for further testing if necessary. Again, authors must provide information on the sample size, and if possible the sample mass. How much is it necessary to obtain an NMR spectra, and how does it compare with the amount required for a microFTIR analysis?

ANSWER: Regarding the analysis of coating, it was not possible to analyse it through microFTIR because this typology of instrument was not available to our laboratory. Taking into account that the coating material had to be cleaned from the art object and, consequently, a big amount of sample (2-3 mg for every cotton swab) was available for the analysis, we decided to perform NMR measurements. NMR experiments, in fact, provide information on chemical structures of most the organic compounds and several identification data can be obtained from the chemical shift without a reference database. Furthermore, in case several species -a mixture of compounds- could be detected in the coating, NMR 2D-experiments would have been performed in order to achieve their structural identification. For this reason, NMR was selected as analytical technique.

Another important point is related with the "3D colour reconstruction proposed in Fig 1B"

From the data presented and discussed in their manuscript, it will be not possible to propose a colour reconstruction. It is not possible to predict a final colour through a pigment; pigments are certainly important for determining colour, but colour built-up & technique are also fundamental for the final colour. Colours were not "uniformly sprayed" in antiquity.

Possibly, in the legend for Figure 1B and in the text, authors could propose colour areas for blue, pink, etc. These areas would be indicative of the generic hue, but should not be considered a colour reconstruction.

However, to be able to provide these generic colour areas, more data than what is provided by micro-sampling and discussed in the paper, is necessary. Possibly, authors had used XRF to indicate areas of colour? and they did not describe it in the manuscript submitted

ANSWER: During the last restoration (2016-2017) on the sarcophagus a campaign was carried out in order to identify the constituent materials. Before of the intervention, the sarcophagus was affected by various degradations and alterations. The chromatic films of the sarcophagus of Hasti Afunei, original or of previous restorations, was quite appreciable, on a macroscopic level, before the beginning of the diagnostic campaign and our subsequent restoration work. Following the restoration, many other chromatic films have been identified. A mapping of the chromatic films present on the surface of the sarcophagus was also carried out. From the large and extraordinary presence of ancient pigment, original or not, it was possible to select some areas for sampling. Once assessed the identification of pigments employed, the authenticity of the work has been verified, distinguishing also the previous restoration interventions. In fact, pigments that are not compatible with the historical period of the archaeological artwork and the presence of surface protections have been identified and related to different period.

Thanks to this information, a 3D reconstruction has been made, once identified all the original pigments employed (Figure 1b). Consequently, the virtual reconstruction of the sarcophagus chromatic backgrounds was carried out by evaluating both microscopically and macroscopically the entire surface in the artifact. In fact, where the presence or residues of pigments were not found, it was not possible to determine a colouring as shown in the virtual model. By diagnostic analysis, archaeological study, comparison with other similar Etruscan works of art it was possible to imagine the original state of the artifact. The result of the crossed data was used to evaluate a reversible and delicate restoration intervention. However, this explanation was added in the text.

Other points, related to A), that is fundamental to fully address are highlighted bellow

1) For Egyptian blue, please describe band assignments (in the text or in a table)

2) Please provide spectra of reference samples, acquired in the same conditions, for goethite and hematite

3) Red pigment in purple, please provide band assignments

4) Green pigment; it is fundamental to provide spectra of reference samples in Fig 7 and it is necessary to discuss in detail its composition; please provide band assignments

5) Pink pigment, please provide reference compound, peak assignments and discuss the accuracy of your prediction in which concerns the identification of a madder lake pigment

Please discuss, why you discard that it could also be a restoration pigment? Considering that in until recently, at least, Winsor & Newton was selling a "genuine" madder lake pigment.

ANSWER: All the descriptions required have been provided. A further explanation of why madder lake has to be considered original and not a restoration pigment has been added in the text.

Reviewer 2 Report

The article entitled „Deepening inside the pictorial layers of the Etruscan sarcophagus of Hasti Afunel: an innovative micro-sampling technique for Raman/SERS analyses” was focused on the interesting aspect of Raman application for the identification of pigments in archeological objects. The aim was very interesting, methodology as well, but the construction of the whole paper is strange and in my opinion incorrect. The greatest doubts are associated with the results part, especially with the subtitles focusing on the analysis of the individual pigments while in reality representing the historical background. In my opinion, in these paragraphs was too less the Raman analysis. Similar problems have resulted from SERS because there is practically no information about the substrates or SERS effect, especially in comparison to the classic Raman effect (lack of data comparing classic and SERS approach). What is more, there is many grammatical and typological errors as well as many repeats causing that the text seems to be written very chaotic. Another problems and doubts are summarized below:

- In the Introduction section, I found incomplete data e.g. SER.. or sentences which are strange e.g. “ Object of this investigation is a….”

- In the section Results: Surface protection results, it seems unclear to me to say "protective elements". What does this mean? What elements do the authors mean? How the elements can be a protective layer? What types of materials are considered as protective? I also think that this part is very poor and should be supplemented with more details.

- In my opinion, there is a lack of selection in the main image (Figure 1) of individual analyzed areas (areas A- ..). This information can be much clearer in receiving for the reader. 

- The analysis and quality of Raman spectra is very poor (eg illegible descriptions in the drawings, not very accurate scale, images in insets are also unreadable, no statistical data suggesting that spectra come from only one place, etc). The description and interpretation of spectra are also very poor and focuses only in some places on the location of bands without analyzing the compound or on the laconic description of the analyzed relationship (eg - the band is associated with the vibration of ..., the bands suggest the presence of ...). More in detail:

  - analysis of "red pigments" is strange because Raman spectra do not univocally indicate the presence of hematite (broadened lines, no lines typical of this phase), while the goethite pattern (FeOH3)? is incorrect. This paragraph, similar to others, is a good example of local analysis. A lot of historical facts, and little Raman analysis. Why were only the A and G samples analyzed, but not the average ones?

  - "black pigment" analysis seems questionable. It seems that the poorly outlined bands correspond to the phases of iron oxides and the bands from the carbon groups may be related to the decomposition of the pigment due to the too high laser power.

  - analysis of "green pigments", how found the presence of particular phases mentioned in the text, eg Ba, Pb, Cu phases? There is no chemical analysis confirming this type of interpretation.

  - "pink pigment" analysis, on which basis an individual bands were analyzed? do you know what kind of compounds this Raman pattern can represent? what does it mean the mysterious GT-SERS abbreviation?

- The "discussion" part is to a large extent the reproduction of data, both on the history and spectrum analysis, as in the case of results part. It seems to me that both parts should be properly rewritten so that in part results should only result, while the discussion should include a summary of the data.

- The "experimental" part. The sentence "referring to [11] ..." should be rewritten because it is difficult to receive. Other problems in the methodical part: Why was the 785nm laser chosen when the samples were fluorescing it? What were the lasers with different wavelength tested for? Why was the 632nm laser used only for "SERS" and not also for classical Raman? What was the laser power on the sample? What was the preparation methodology of the substrate for SERS? How is it known that this technique was used? (no data comparing results from classic Raman and SERS: is it possible to collect such set of data?). If, however, failed to measure the classic Raman, why did SERS succeed? - it is so strange that SERS is a reinforcement technique which should raise the fluorescence effect. What types of compounds generate fluorescence and why (the presence of conjugated bonds?).

Author Response

Reviewer #2:

Comments to the Author

The article entitled „Deepening inside the pictorial layers of the Etruscan sarcophagus of Hasti Afunel: an innovative micro-sampling technique for Raman/SERS analyses” was focused on the interesting aspect of Raman application for the identification of pigments in archeological objects. The aim was very interesting, methodology as well, but the construction of the whole paper is strange and in my opinion incorrect. The greatest doubts are associated with the results part, especially with the subtitles focusing on the analysis of the individual pigments while in reality representing the historical background. In my opinion, in these paragraphs was too less the Raman analysis. Similar problems have resulted from SERS because there is practically no information about the substrates or SERS effect, especially in comparison to the classic Raman effect (lack of data comparing classic and SERS approach).

ANSWER: The results part has been completed rewritten and the discussion about SERS spectra have been provided. A comparison between Raman and SERS effect has not provided because, as reported in the paper, the Raman analyses with Raman portable instrument did not provided any reliable signals, due to high fluorescence background caused by coating wax. Furthermore, this phenomenon characterized also organic compounds and we do not report a comparison between Raman and SERS, such as madder lakes, because it is amply studied and reported in different papers, such as MicroRaman Spectroscopy (MRS) and Surface Enhanced Raman Scattering (SERS) on organic colorants in archaeological pigments, by Elsa Van Elslande, Sophie Lecomte and Anne‐Solenn Le Hô, in Journal of Raman Spectroscopy, 2008, 39, 1001-1006.

What is more, there is many grammatical and typological errors as well as many repeats causing that the text seems to be written very chaotic. Another problems and doubts are summarized below:

- In the Introduction section, I found incomplete data e.g. SER.. or sentences which are strange e.g. “ Object of this investigation is a….”

- In the section Results: Surface protection results, it seems unclear to me to say "protective elements". What does this mean? What elements do the authors mean? How the elements can be a protective layer? What types of materials are considered as protective? I also think that this part is very poor and should be supplemented with more details.

ANSWER: First of all, all the text has been revised according to the suggestion of improving English language. Furthermore, we would underline that the word “elements” could result ambiguous: we meant “elements” as a generic expression for “agent” and not referring to “chemical elements”. For this reason, we substituted the word “elements” with “coating agents”, which results more coherent with the identification of wax as protective layer.

In my opinion, there is a lack of selection in the main image (Figure 1) of individual analyzed areas (areas A- ..). This information can be much clearer in receiving for the reader. 

- The analysis and quality of Raman spectra is very poor (eg illegible descriptions in the drawings, not very accurate scale, images in insets are also unreadable, no statistical data suggesting that spectra come from only one place, etc). The description and interpretation of spectra are also very poor and focuses only in some places on the location of bands without analyzing the compound or on the laconic description of the analyzed relationship (eg - the band is associated with the vibration of ..., the bands suggest the presence of ...). More in detail:

  - analysis of "red pigments" is strange because Raman spectra do not univocally indicate the presence of hematite (broadened lines, no lines typical of this phase), while the goethite pattern (FeOH3)? is incorrect. This paragraph, similar to others, is a good example of local analysis. A lot of historical facts, and little Raman analysis. Why were only the A and G samples analyzed, but not the average ones?

  - "black pigment" analysis seems questionable. It seems that the poorly outlined bands correspond to the phases of iron oxides and the bands from the carbon groups may be related to the decomposition of the pigment due to the too high laser power.

ANSWER: differences between SERS spectra acquired on gel and Raman reference has been explained in the text. The decomposition of the pigment due to high laser power is unlikely, because preliminary measurement at very low intensity of the laser and with short acquisition time showed the presence of these bands, so it is highly probable carbon black was used as pictorial pigment (this use is attested in literature).

-analysis of "green pigments", how found the presence of particular phases mentioned in the text, eg Ba, Pb, Cu phases? There is no chemical analysis confirming this type of interpretation.

  - "pink pigment" analysis, on which basis an individual bands were analyzed? do you know what kind of compounds this Raman pattern can represent? what does it mean the mysterious GT-SERS abbreviation?

- The "discussion" part is to a large extent the reproduction of data, both on the history and spectrum analysis, as in the case of results part. It seems to me that both parts should be properly rewritten so that in part results should only result, while the discussion should include a summary of the data.

ANSWER: indication has been followed and all the details referable to pink pigment have been put in the text

The "experimental" part. The sentence "referring to [11] ..." should be rewritten because it is difficult to receive. Other problems in the methodical part: Why was the 785nm laser chosen when the samples were fluorescing it? What were the lasers with different wavelength tested for? Why was the 632nm laser used only for "SERS" and not also for classical Raman? What was the laser power on the sample? What was the preparation methodology of the substrate for SERS? How is it known that this technique was used? (no data comparing results from classic Raman and SERS: is it possible to collect such set of data?). If, however, failed to measure the classic Raman, why did SERS succeed? - it is so strange that SERS is a reinforcement technique which should raise the fluorescence effect. What types of compounds generate fluorescence and why (the presence of conjugated bonds?).

ANSWER: The first sentence in “Experimental” was rewritten.

About the adopted instruments, it is important to evidence that, unfortunately, the available Raman spectrometers used for the analyses present different wavelength lasers, so no choice and no comparison was really possible. In particular, for the on-site Raman analysis, the 785 nm laser instrument was used and this usually guarantees the acquisition of spectra less affected by fluorescence: the laser wavelength is close to InfraRed, so this laser source is favorite for Raman analysis on samples where no preliminary composition information is available, as in this case. Unfortunately, because of the high content of aged compounds (e.g.: the pictorial binder), the 785 nm laser did not result sufficient for the acquisition of informative spectra. Regarding the analysis of samples on gel matrix, the instrument presents only a 632 laser. In these cases, it is not possible to distinguish between Raman and SERS analysis, because they are based on the same physical phenomenon (Raman Scattering) which, in presence of metal nanoparticles and direct contact between them and analytes molecules could result in signal Enhancement (SERS): the gel matrices contain silver nanoparticles inside, so it is not possible to distinguish between standard Raman and SERS. This is evidenced by the presence of characteristic Ag colloid in every spectrum (band evidenced in the text). However, this results in advantages for the versatility of the kit: this one can be applied on pictorial films without any preliminary information on their composition, granting for acquisition of spectra both from compounds which do not show particular issues for Raman detection (e.g. Haematite) and species with lower detectability at standard Raman but great SERS affinity (e.g. organic dyes). In order to provide a better explanation of these aspects, we modified drastically the text adding further details and clarifications.

Regarding the preparation of the gel substrate, it was obtained following the experimental protocol reported in the cited patent by one of the author of the same: the characterization test (microscopy observation, UV-Vis Absorption spectroscopy, Raman analysis of the matrix blank) were performed and they confirmed the idoneous characteristics of the substrate.

Reviewer 3 Report

The paper deals with the innovative microsampling technique for Raman/SERS analyses in the study of pictorial layers of Etruscan sarcophagus of Hasti Afunei. The results obtained in this work are interesting enough. I recommend to reconsider after major revision. Comments and suggestions listed below:

Introduction: In general, I think that it is too brief. You could revise the state-of-art of SERS application adding new modern references. 

Results: please use the same name for the figures (Fig. or Figure). 

Figures displaying spectra are of poor quality. Please prepare quality figures for publication. 

Author Response

Reviewer #3:

Comments to the Author

The paper deals with the innovative microsampling technique for Raman/SERS analyses in the study of pictorial layers of Etruscan sarcophagus of Hasti Afunei. The results obtained in this work are interesting enough. I recommend reconsidering after major revision. Comments and suggestions listed below:

Introduction: In general, I think that it is too brief. You could revise the state-of-art of SERS application adding new modern references. 

Results: please use the same name for the figures (Fig. or Figure). 

Figures displaying spectra are of poor quality. Please prepare quality figures for publication. 

ANSWER: all the suggestions have been accepted in the text.

Round 2

Reviewer 2 Report

I would like for addressing my comments and doubts. Now, the text is much much better and present data more clearly. However, not all my questions were taken into consideration e.g.:

  - Why were only the A and G samples analyzed, but not the average ones?

-analysis of "green pigments", how found the presence of particular phases mentioned in the text, eg Ba, Pb, Cu phases? There is no chemical analysis confirming this type of interpretation.

I have also question about the silver particles. What kind of silver (metallic, oxidized, spherical, aspherical, size, homogeneity of size distribution) were applied? Are the characteristics of silver were checked before matrix preparation? Could you fulfill the text about the information associated with silver and its characteristics? It is important in the case of the SERS effect.

Please also homogenize the scale on the axes and increase the font size in the pictures

In many parts of the text, I have still a problem with English grammar. Please correct the language maybe by the help of native speaker and make it more fluent in reading.

Author Response

Reviewer #2:

Comments to the Author

I would like for addressing my comments and doubts. Now, the text is much much better and present data more clearly. However, not all my questions were taken into consideration e.g.:

Why were only the A and G samples analyzed, but not the average ones? ANSWER: All the samples, taken with a scalpel in correspondence of the points reported in Figure 2, were analysed with Raman spectroscopy. All the spectra related to the same color resulted evidently consistent among each other in spectral features, as cleared in the text. Furthermore, the area where the samples were taken are not close and belong to different details of the art object, with no evident correlation with the exception of the same colour. For all these reasons, an average spectrum would have not provided any particular additional information, so this typology of data treatment was evaluated not necessary and not useful for the final information content. The spectra reported in figs. 4 and 5 are provided as illustrative of the spectral features common to points of the same color. To make it clearer, this was evidenced in the text (green marked).

Analysis of "green pigments", how found the presence of particular phases mentioned in the text, eg Ba, Pb, Cu phases? There is no chemical analysis confirming this type of interpretation. ANSWER: The Raman analyses provide molecular structure information, so the identification of Barium, Lead and Copper pigments mentioned in the text is based on the experimental Raman shift of peaks and their comparison with the literature, as reported in the text. These peaks are indicative of the specific compounds and not of the generic elements that could be present, even if, as mentioned in the text, the assignment is only tentative due to the low signal to noise ratio. No further chemical analyses for the identification of these elements in the green area was performed. To evidence this uncertainty, we also added this clarification in the text (green marked).

I have also question about the silver particles. What kind of silver (metallic, oxidized, spherical, aspherical, size, homogeneity of size distribution) were applied? Are the characteristics of silver were checked before matrix preparation? Could you fulfill the text about the information associated with silver and its characteristics? It is important in the case of the SERS effect.

ANSWER: The silver nanoparticles are metallic and obtained by reduction of Ag precursor. The silver nanoparticles are characterized by UV-Vis spectroscopy and microscopy techniques before their insertion in gel matrix. All the information is reported in the patent mentioned in the text, which constitutes the main reference for the related data and the main source for their fruition.

Please also homogenize the scale on the axes and increase the font size in the pictures

ANSWER: We modified the figures, which have now the same dimension and scales; the font size was increased. No modification was done for the range of the spectra, which highlights the most important peaks.

In many parts of the text, I have still a problem with English grammar. Please correct the language maybe by the help of native speaker and make it more fluent in reading. ANSWER: We have adopted your suggestion, revising the English.

Reviewer 3 Report

In my opinion the paper can be accept in present form.

Author Response

Reviewer #3:

Comments to the Author

In my opinion the paper can be accept in present form.

ANSWER: Thank you for your comment.